# Switching from Cigarettes to Heated Tobacco Products in Japan—Potential Impact on Health Outcomes and Associated Health Care Costs

**DOI:** 10.3390/healthcare12191937

**Published:** 2024-09-27

**Authors:** Joerg Mahlich, Isao Kamae

**Affiliations:** 1Department of Economics, University of Vienna, Oskar Morgenstern Platz 1, 1090 Vienna, Austria; joerg.mahlich@univie.ac.at; 2Düsseldorf Institute for Competition Economics (DICE), University of Düsseldorf, Universitätsstr. 1, 40225 Düsseldorf, Germany; 3Graduate School of Public Policy, The University of Tokyo, 7-3-1 Hongo, Bunkyo-ku, Tokyo 113-0033, Japan

**Keywords:** heated tobacco products, Japan, health resource utilization, smoking

## Abstract

Background: Japan’s rising health expenditure, driven by an aging population, coincides with growing demands for increased spending. Reducing smoking-related costs could alleviate the burden on the health care system. Despite efforts to promote smoking cessation, success has been limited, indicating a need for strategies beyond cessation. Methods: Using a status quo simulation based on hospital resource data from the Japanese Ministry of Health, Labor, and Welfare, we examine the impact of heated tobacco products (HTPs) on the prevalence of four smoking-attributable diseases (chronic obstructive pulmonary disease, ischemic heart disease, stroke, and lung cancer) and the related direct health care costs. The baseline scenario assumes a 50% switch from combustible cigarettes to HTPs, with a 70% risk reduction. A sensitivity analysis was conducted to assess the effects of parameter variations. Results: If 50% of smokers replaced combustible tobacco products with HTPs, 12 million patients could be averted equivalent to JPY 454 billion in health care savings. Prefectures located in the north and south of Japan would benefit the most. Conclusions: Considering the heterogeneous prevalence rates, a one-size-fits-all tobacco control approach is ineffective. Japan should prioritize cost-efficient measures that promote public health and economic benefits. Encouraging smokers to switch to reduced-risk products, raising awareness of health risks, and adopting a harm-based taxation model can drive positive change. Public–private partnerships can further enhance harm reduction efforts. With a combination of tax reforms, revised regulations, collaborations, and ongoing research, Japan can create a more effective and comprehensive approach to tobacco control.

## 1. Introduction

Japan’s health care system has been very strong, delivering long life expectancy at a relatively low cost [1]. However, costs have risen in recent years and are projected to increase further due to expensive medical advances and the aging population [2]. Japanese citizens are currently forecast to live to the age of 82 for men and 88 for women as of 2020, increasing to 85 for men and 91 for women by 2060 [3]. Seniors do not necessarily burden the economy though—if fit and healthy, elderly adults represent a valuable and often-utilized asset, for example for additional labor or childcare assistance [4]. However, health care spending scales up when seniors become ill, especially since this increases the likelihood of them requiring social care, for example through nursing homes [5,6].

Although the Japanese health sector performs well above the Organization for Economic Cooperation and Development (OECD) average, several factors put pressure on hospital resources. These include higher risk factor-related mortality and increased mortality from aging-related conditions. While increasing health expenditures were not a major concern in times of economic growth, the stagnating economy and increasing inflation challenge the public as well as private medical care sectors. The pressure on hospital management intensifies, adding to the existing issues of skilled labor force shortages and high mental loads of medical professionals. Thus, Japanese hospitals need to become more cost efficient and reduce the number of patient admissions [7]. To improve health outcomes, a reduction in behavioral risks such as alcohol abuse and smoking should be prioritized. Minimizing smoking-related risks should be a cost-effective measure, and initial success has already been shown in Japan. This demand has been already acknowledged by the government and health professionals [8,9]. The Japan Vision: Health Care 2035 included the roadmap to a “Tobacco-free” Tokyo Olympics 2020 and a “Tobacco-free” society by 2035, suggesting policy options predominantly aiming for prevention and cessation either through tobacco tax increase, packaging and advertising regulations, smoking cessation support, or treatments [8]. A global review of tobacco policies reveals that most governments aim for similar approaches [10]. There is a large body of literature that has investigated the effectiveness of such measures. A study in Korea showed that, while a national smoking cessation program reduced pneumonia costs among patients below 50 years, it raised overall health care costs [11]. In France, providing free access to smoking cessation treatments was found to cost at least EUR 125 million initially and save at minimum EUR 15 million over five years [12]. Similarly, a U.S.-based model predicted that implementing comprehensive smoking cessation policies, such as expanded treatment coverage and the promotion of quit lines, may reduce smoking prevalence by 3 percentage points within one year [13]. A study conducted by Igarashi et al. (2016) simulated the impact of a medicated smoking cessation aid using observations obtained from varenicline [14]. They found that varenicline was able to reduce smoking-attributable health care costs by only 6%.

Switching from traditional cigarettes to HTPs or e-cigarettes has been explored to some extent as a potential support or alternative to smoking cessation, but research remains limited, and the findings have been mixed. Some studies indicate that smokers switching to HTPs showed significant reductions in exposure to harmful substances compared with those continuing to smoke cigarettes [15]. Research on patients with chronic obstructive pulmonary disease (COPD) shows fewer exacerbations and improvements in walking distance [16]. A longer-term study also supports these findings, showing that smokers who switched to HTPs experienced sustained reductions in exposure to toxicants over a year, positioning HTPs as a viable option for smokers who do not intend to quit smoking [17].

The secondary risks of harmful consumption, such as the interlink between cigarette smoking and increased alcohol consumption or reduced physical activity, might also be affected when switching to HTPs, though these effects are still underinvestigated [18,19]. Switching to HTPs could help in the transition to a more health-conscious lifestyle, e.g., HTPs are associated with reduced oxidative stress compared with cigarette smoking, potentially lowering barriers to engaging in more physical activities [20]. Furthermore, exclusive HTP users tend to consume fewer sticks per day than cigarette smokers do, with an average of 10 sticks per day compared with 15 cigarettes reflecting a more mindful approach to consumption [21].

In Japan, while up to 77% of male adults smoked cigarettes in the 1950s, the overall prevalence declined to 17% in 2019 [22,23]. This may be due to the introduction of strict measures to reduce smoking in public places in combination with the Japanese enthusiasm for technological innovations. Japan was one of the earliest markets in which alternatives to traditional combustible cigarettes were made available in the form of heated tobacco products (HTPs). Given its singularity in Japan—the government has not yet legalized any other similar non-combustible product, such as snus, nicotine pouches, or liquid alternatives—the Japanese HTP market can be considered as a model for analyzing the comparative risks related to HTP or combustible smoking.

Four HTP producers compete in the Japanese market with four different technology platforms, which fosters innovation in the Japanese HTP market [24]. Despite this market evolvement on the supply side, demand has slowed down in the past years. In 2019, the prevalence of HTP use was 3%. Of all tobacco users, only 7% of males and 5% of females consume both cigarettes and HTPs [23]. Both the low share of dual use and a rapidly declining smoking prevalence since HTP introduction point to the substitution potential of HTPs. This hypothesis is backed by a Japanese study finding that HTPs contributed to a reduction in cigarette sales [25]. Nevertheless, the smoking prevalence was still at 17% in 2019, causing a significant number of preventable disease incidents. Reducing the smoking prevalence would help mitigate pressure on health costs and hospital resources. Measures to target specific subgroups may be essential, especially for those that still resist the overall trend toward smoking cessation. For example, the smoking prevalence was 27% among male adults in 2019, and, among low-educated men and women, it was 58% and 35%, respectively, in 2016 [26]. In addition, smoking behavior differs substantially between regions, potentially being correlated with economic strength, demography, and cultural differences. The highest smoking prevalence rates are found in the regions of Hokkaido (27% in 2016) and Tohoku (21%), which are in the north of Japan, as well as in Chugoku (21%), Shikoku (21%), and Kyushu (21%), which are in the southwest [23].

The objective of this study is to address the existing literature gap concerning the macroeconomic consequences of the health implications of HTPs. Much of the current research on HTPs is concerned with the health effects of consumption at the individual level, with a particular focus on the behavioral and physiological impacts at the microlevel [25,27,28,29,30,31,32,33,34,35,36]. However, there has been comparatively little exploration into the influence of these estimated health outcomes on macroeconomic factors. A modest but growing body of research has begun to address this issue, with similar simulations conducted for the United States (U.S.) and Mexico for e-cigarettes [37,38]. This study will be the first to apply such simulations in the context of Japan, as well as the first to do so specifically for HTPs. Notably, related simulations of the tobacco interventions in Japan, such as those modeling the impact of varenicline for smoking cessation [14], will be used for comparison in our analysis.

Japan’s high smoking prevalence, combined with its widespread acceptance of smoke-free alternatives, presents a unique opportunity to investigate whether transitioning smokers to heated tobacco products (HTPs) could alleviate the burden on the health care system. We seek to address the question: Can risk-reduced products serve as a viable alternative for mitigating the health burden associated with smoking?

## 2. Materials and Methods

This research consisted of three steps. First, based on a literature review, we established the association between cigarette smoking and smoking-induced diseases, such as lung cancer, cardiovascular diseases, or chronic obstructive pulmonary disease (COPD), and health care cost. Then, based on a literature review, we determined the impact of switching from cigarettes to HTPs on the associated health hazard. This literature included published epidemiological data and studies on the potential of HTPs. Finally, changes in the risks were translated into corresponding changes in health outcomes, survival rates, and associated changes in health care costs. The model calculations are presented in Appendix A.

### 2.1. Smoking-Attributable Function

A literature review was conducted on the disease burden related to smoking and the reduced disease burden associated with HTP use instead of smoking. To estimate the burden of disease of smoking, epidemiologists rely on the population-attributable fraction (PAF), which is the country-specific proportion of incidents attributable to a certain risk factor. As the risk factor of interest is smoking, the PAF can then be called the smoking-attributable fraction (SAF). To determine the number of patients who developed a disease due to smoking, we used disease-specific relative risks from meta-studies by Thun et al. (2000) and Gandini et al. (2008) [39,40]. Thus, we did not rely on a single source for relative risks, which tend to vary over studies. To estimate Japan’s population-attributable fractions, we followed the approach used in “The Preventable Risk Integrated Model” a World Health Organization (WHO)-supplied tool from Scarborough et al. (2016) [41]. For this, we calculated the number of patients by smoking status and disease, weighted by their respective relative risks [39,40].

### 2.2. Risk Reduction Potential of HTPs

Published evidence was used to determine the effect of so-called reduced-risk products, particularly HTPs. A study by Forster et al. (2017) indicates an absolute risk reduction of 97% [27]. The authors investigated the nine toxicants proposed by the WHO Study Group on Tobacco Product Regulation (TobReg) for mandated reduction in cigarette emissions. They found an overall average reduction of 97.1%. Li et al. (2019) estimated 80% fewer harmful constituents in the releases from HTPs [28]. There are several studies that found that the content of HTP smoke contained 70–95% of the concentration of nicotine and toxicant exposure found in traditional cigarette smoke [25,29,30,31,32,36,42]. Another study conducted by Nutt et al. (2014) employed a rather unusual method: a multicriteria decision analysis model of the relative importance of different types of harm related to the use of nicotine-containing products [33]. This expert-based analysis concluded that the risk related to the use of HTPs is 96% lower than that for cigarettes. The study by Zhang et al. (2023) showed significantly lower exhaled levels of carbon monoxide with HTPs than that with cigarettes [34]. The lower toxicant levels were confirmed by Bekki et al. (2017) of the National Institute of Public Health Japan who found that HTP smoke contained one hundredth of the amount of carbon monoxide compared with combustible cigarette smoke [35].

### 2.3. Translating Reduced Emissions into Health Impacts

Given the recent introduction of HTPs, there is no longitudinal epidemiologic study allowing the analysis of the impact of reduced emissions on individual or public health. However, public health researchers suggest that beneficial health impacts are plausible. The Dutch National Institute for Public Health and the Environment (RIVM) has derived the change in cumulative exposure (CCE) of HTPs and cigarettes, which was then translated into an estimate of the health impact—a change in expected life span [43]. Assuming an absolute risk reduction of 90%, the life expectancy of HTP consumers would be closer to the life expectancy of non-smokers than to that of smokers. In a detailed modeling assessment, Stephens (2017) compared the relative harmfulness of different nicotine products with a model based on exposure data and cancer potencies [44]. The calculated lifetime cancer risk of the HTP was one to two orders of magnitude lower than that for combustible cigarettes. The meta-studies of Znyk et al. (2021), Uphadyay et al. (2023), and Yayan et al. (2024) provide literature reviews on the impact of reduced-risk products on individual and public health [45,46,47]. Among other health impacts, Znyk et al. (2021) found that HTP use is correlated with a decreased risk of lung cancer and cardiovascular diseases. Yayan et al. (2024) noted that respiratory damage caused by e-cigarettes and cardiovascular health consequences could be less severe than those related to conventional cigarettes.

### 2.4. Model Input

Based on the literature review outlined above, we populated our model with the parameters reported in Table 1. We used the latest available prefecture-level smoking prevalence data from the 2016 National Health and Nutrition Survey (NHNS) [23,42]. To estimate smoking prevalence for 2022 at the regional level, we extrapolated the data using the compound annual growth rate (CAGR) of national smoking prevalence observed between 2016 and 2019. This method assumes that the trend observed at the national level reflects similar regional trends, allowing for the estimation of 2022 values even in the absence of region-specific data for the years after 2016. By applying the CAGR to each prefecture’s 2016 smoking prevalence, we calculated the projected prevalence for 2022.

The current number of inpatient and outpatient visits and the associated direct health care costs for chronic obstructive pulmonary disease (COPD), ischemic heart disease (IHD), stroke, and lung cancer were employed from the Handbook of Health and Welfare Statistics, which is also established by the Ministry of Health, Labor, and Welfare [48]. The considered health care costs included the inpatient and outpatient visit costs, as well as the medical care expenditures for preparations from the pharmacy and medical care expenditures in general. Two parameters are critical for estimating the effect of switching from cigarettes to HTPs: the switching rate and the degree of risk reduction. The risk reduction for smokers switching to HTPs was set to 70% in the base-case scenario. The base-case scenario assumes a switch rate of 50%, i.e., one half of smokers switch to HTPs while the other half continue using traditional tobacco products.

For the cost, the results are reported for Japan and on a prefectural level. Given a residual excess health risk in the initial period when smokers quit, we applied a conservative health risk reduction. 

At a later stage, we also conducted a deterministic sensitivity analysis to accommodate the uncertainty around some of the model parameters. Specifically, we varied the degree of risk reduction (50% and 90%), the switching rate to HTPs (25% and 75%), the smoking prevalence (10.4% and 20.4%), and the health care costs (inpatient and outpatient visit costs vs. total health care costs excluding expenses for dental care) (Table 2).

## 3. Results

Reported in Table 3, the annual number of smoking-attributable patients would be reduced by more than 12 million cases (26%) in the baseline scenario. Stroke cases as well as IHD cases would be reduced by 29%, i.e., more than seven million fewer stroke cases and more than two million fewer IHD cases than those in the status quo (Figure 1).

The savings in associated health care costs are summarized in Table 4. According to our estimates, the total health care costs for smoking-attributable diseases could decline from JPY 1778 bn to JPY 1324 bn (a 26% reduction). The highest savings of approximately JPY 247 bn (a 29% saving) could be generated by the reduced numbers of stroke cases (Figure 2).

The regional breakdown of the savings is reported in Figure 3. To account for differences in population size, the cost savings are displayed in million JPY per 100,000 people, with darker colors representing increased savings. The prefectures located in the north and south of Japan could benefit significantly more from smokers switching to HTPs than the middle and eastern prefectures.

Figure 4 reports the results of the sensitivity analysis for the expected cost savings. The switching rate and the risk reduction are the two parameters that influence the results of the baseline scenario the most. With a risk reduction of 90%, cost savings could amount to JPY 607 bn. Similarly, savings of JPY 730 bn could be achieved with a switching rate of 75% and base-case risk reduction. For health care costs, we used the costs for inpatient and outpatient visits as our lower bound and the total health care costs excluding dental care as our upper bound. A variation in the smoking prevalence would influence the results of the baseline scenario marginally (Figure 4).

To validate our results, we compared our calculated smoking-attributable fractions with the ones measured by the Global Burden of Disease Study and Katanoda et al. (2008) [51,52]. Our analysis indicated an SAF for lung cancer of 62%, which aligns closely with the intermediate values between 57% (GBD) and 69% (Katanoda et al.). Similarly, for IHD, our findings indicated an SAF of 35%, adjoining figures from the GBD at 15% and Katanoda et al. at 44%. Notably, our stroke SAF of 36% surpassed the corresponding SAFs 16% (GBD) and 10% (Katanoda et al.), and our COPD estimate of 68% modestly exceeded the 60% reported by Katanoda et al. These comparisons suggest that our findings generally reside within the mid to upper range of the existing estimates for SAFs. Varying results might occur due to the application of different sources for smoking prevalence and relative risks. Our calculations on the smoking-attributable costs as a share of the total health care spendings of 4.4% are close to those of Izumi et al. (2001), who reported 3.8%. Although these comparisons support the robustness of our findings, the next step is to explore how these insights can be effectively applied within the context of Japan’s health care challenges.

## 4. Discussion

The Japanese health care system has long been acclaimed for its performance in terms of delivering the world’s longest average life expectancy at a relatively low cost [53]. However, in recent years, Japan has been struggling with rising health care costs, mainly due to its “super ageing” population [54]. Even now, hospital doctors suffer from overwork and burnouts [55]. Considering a further increase in life expectancy in future decades, health costs are expected to rise significantly. One way for the Japanese government to decrease the share of its GDP that is spent on health care is to reduce the risks related to smoking. Switching smokers to HTPs seem to offer an effective means for reducing health care risks and related health care cost.

Similar to that in other countries, the public health policy in Japan traditionally focuses on smoking cessation, and cessation treatments have been covered under universal health insurance in Japan since 2006. In April 2006, Chuikyo (Central Social Insurance Medical Council), which is an advisory body to the Minister of Health, Labor, and Welfare that deliberates on revisions to Japan’s health insurance system and medical fees, included outpatient smoking cessation guidance (nicotine dependence management fee) in the National Formulary. This was the first time in history that the Chuikyo decided on insurance coverage based on cost-effectiveness considerations. The following four points were discussed at that time: (1) verification of cost effectiveness was required for insurance coverage; (2) domestic economic evaluation research was required; (3) it was not about reducing costs, and good cost effectiveness was accepted; and (4) model analysis was not regarded as evidence [56]. However, the discussions within the government stagnated regarding the cost-effectiveness evaluation until the trial implementation of cost-effectiveness evaluation in 2016 [57].

One recent example is the CureappSC digital smoking cessation app, which received reimbursement in November 2020 [58]. The app is used together with a portable device that measures the carbon monoxide concentration in a patient’s breath [59]. However, actual adoption is limited in Japan [60]. Even though the public health policy encourages cessation on many communication levels, this approach has seen limited success.

One reason is that many smokers do not necessarily find smoking cessation desirable even if they are aware of the risk smoking imposes on health. In Japan, only one quarter of smokers wants to quit [23]. The barriers to the use of cessation services are the fear of being judged or a fear of failure, which might lead to too much pressure to even make an attempt to quit [61]. Many smokers are, therefore, more open to reducing smoking than to fully quitting because this flexibility allows them to maintain their autonomy, giving them a sense of control over their smoking behavior. Nevertheless, evidence shows that smokers who reduce the number of daily cigarettes smoked are more likely to attempt and achieve smoking cessation, making smoking reduction an intervention tool [62]. However, with no further support or replacement treatment, approximately 86% of those that report quitting smoking relapse back into their harmful behavior. Exclusive policy focus on cessation, therefore, may not be a sustainable strategy, and other, new solutions need to be found to reduce smoking-related disease burdens—such as facilitating a switch to risk-reduced products such as HTPs. Sweden, as an example, has the lowest rate of tobacco-related disease in Europe and is the only EU state to have reached smoke-free status with less than 5% adult smoking prevalence. This could be since Sweden offers snus as a smoke-free alternative [63]. Sweden has maintained a unique position in the European Union, having secured an exemption to continue selling snus when it joined the EU, despite the product being banned in other EU countries since 1992 [64]. In 2024, Sweden introduced a 9% tax rise on cigarettes while reducing the tax on snus by 20%, reflecting the focus on harm reduction [65].

Our model indicates a reduction in the incidence of smoking-attributable diseases and a related reduction in cost. According to our baseline estimates, a 50% switch of smokers to HTPs could reduce the number of smoking-attributable diseases such as lung cancer or heart diseases by 12 million patients annually. The associated savings in direct health care costs would amount to JPY 454 bn a year (USD 3 billion) or a saving of 26%.

The impact estimated through our model for the Japanese context compares well with other reports. Igarashi A, et al. (2016) estimate 6% lower medical costs if the smoke cessation drug varenicline would increasingly be used for cessation [14]. In comparison, our model predicts an HTP-related health care cost reduction of 26%. The difference can be explained by the low cessation success rate of varenicline, of which only 14% of individuals undergoing a quit attempt succeed [66]. Our estimated reduction in health outcomes for Japan also points to similar findings when compared with studies conducted in other countries [37,38,67,68].

At a regional level, the highest potential savings could be realized in the prefectures in the South and West of Japan, for instance within Shikoku or Kyushu islands (e.g., Kochi and Kagoshima). These rather rural areas have a lower regional gross product but a high smoking prevalence. Physician shortages in those remote areas are of increasing concern [69,70]. Reducing physician utilization by reducing health risks may be one effective remedy. Considering the heterogeneous prevalence rates across population groups and the uneven age and working population distribution across prefectures, a one-size-fits-all, centralized approach to tobacco control may not be the best answer for the Japanese government. To leave no one behind, health authorities need to employ multiple tools to reduce the negative health impacts of smoking—including both cessation and behavioral change with a device innovation such as switching to HTPs.

## 5. Policy Implications

Implementing cost-efficient measures beneficial to public health and the economy should be the primary goal of the Japanese government. Despite strict regulations, smoking behaviors have not been entirely eliminated, indicating that the current regulatory and fiscal policies have limitations. Motivating smokers to switch from smoking traditional cigarettes to using reduced-risk products and preventing young adults from initiating smoking may be a complementary and effective policy to reduce the negative health consequences and health care costs. In addition, raising awareness of the health risks associated with each product enables the consumer to make an informed, independent, and rational choice.

Currently, the Japanese tax policy follows a rather simple approach by increasing tobacco taxes on all products to raise a minimum amount of revenue, regardless of the existing differences among those tobacco products [71,72]. Thus, the tax authorities do not sufficiently utilize the tax instrument to guide consumers toward less harmful behavior while increasing their tax revenues.

A more effective practice, which is also used in other jurisdictions internationally, would impose higher taxes on the products with the most health risks, i.e., traditional cigarettes, while lowering taxes on products with relatively lower health risks, i.e., HTPs [73]. This may persuade smokers to replace cigarettes with HTPs as they would be more affordable. A harm-based taxation model, in which taxes are aligned with the health risks of each product, balances public health benefits and tax generation.

In addition to tax reforms, the government should reassess its regulations. Currently, restrictions often apply to all tobacco products, including HTPs, despite the lower exposure produced by these products. A balanced approach could incentivize smokers to use less harmful alternatives.

Forming public–private partnerships could further enhance the positive health-directed efforts of the Japanese government. By collaborating with the health sector and employers, the government could leverage wider expertise to promote smoking cessation, also thanks to the transition to HTPs. For example, the health sector could offer smoking cessation support and increase awareness on less harmful alternatives, while companies could use their unique tools to encourage smokers to quit—for example, by offering more days off for non-smokers [74]. Examples of such partnerships already exist in Japan, but, if given a higher priority and including less harmful alternatives, public and private sector goals could be better aligned to make significant progress in reducing the damage caused by risky behavior. Ultimately, companies have an interest in reducing smoking, as smoking reduces productivity due to increased absenteeism (i.e., more sick days) and presenteeism rates (i.e., lower productivity at work) from smoking-attributable disability. Forming public–private partnerships could be a way to achieve that goal.

Finally, investing in studies remains essential for fully understanding the health risks associated with HTPs. Continuous research will provide a virtuous feedback loop of evidence-informed regulatory decisions, ensuring that future policies are both effective and adaptable to evolving tobacco use trends. Through a combination of tax reforms, revised regulations, public–private collaborations, and continued research, Japan can develop a more effective and comprehensive approach to tobacco control.

## 6. Limitations

The simulation rests on several assumptions that are central to the model outcomes. Mainly, we assume that HTPs are a substitute for traditional tobacco products. Although this assumption is backed by the literature, the results would change if HTPs would attract non-smokers. In that case, the population risks would increase rather than decrease with unintentional consequences for health outcomes and costs. In addition, the long-term risks of HTPs have not yet been fully assessed. While the applied rate of 70% risk reduction seems reasonable, some long-term harms might be underestimated. Another assumption is that the reduction in toxicological risks as shown in several studies represents a meaningful reduction in effective health risks for the users. However, we might have also underestimated some of the benefits of switching to HTPs. For instance, a Japanese study conducted during COVD-19 found out that, compared with cigarette-only users, HTP-only users were more likely to quit smoking all together [75].

Another reason why our study might underestimate the benefits from switching is that only four smoking-attributable diseases were considered. Smoking also increases the risk for many other illnesses such as psoriasis, rheumatoid arthritis, or inflammatory bowel disease, which were not covered in our analysis [76,77,78]. Furthermore, only direct health care costs were taken into consideration for our cost estimates. However, indirect costs such as productivity losses are even more relevant. Globally, the total burden of smoking was estimated to be USD 1852 billion (in purchasing power parity) in 2012. Only USD 467 billion (25%) of this amount can be associated with health care expenditures [79].

## 7. Conclusions

This pioneering analysis conducted for Japan has demonstrated that the transition of smokers to HTPs can markedly diminish the health care burden associated with smoking, with an annual savings of 26% or JPY 454 billion and the prevention of 12 million patients, comprising both inpatients and outpatients. Consequently, the adoption of less harmful alternatives for smokers could be incorporated as an additional strategy in tobacco policy design.

This research complements existing studies that have focused on smoking cessation programs and their effectiveness. For instance, the use of medication-based cessation methods, such as varenicline, has proven only marginally effective with a 6% reduction in health care costs [14]. Cessation programs, due to their limited success rates, are insufficient on their own. Future policy design should address the fact that people are generally resistant to outright bans, even if only implied. Japan is an optimal country for policy design that uses a differential approach on different tobacco products, given its already high acceptance rates of HTPs.

## Figures and Tables

**Figure 1 healthcare-12-01937-f001:**
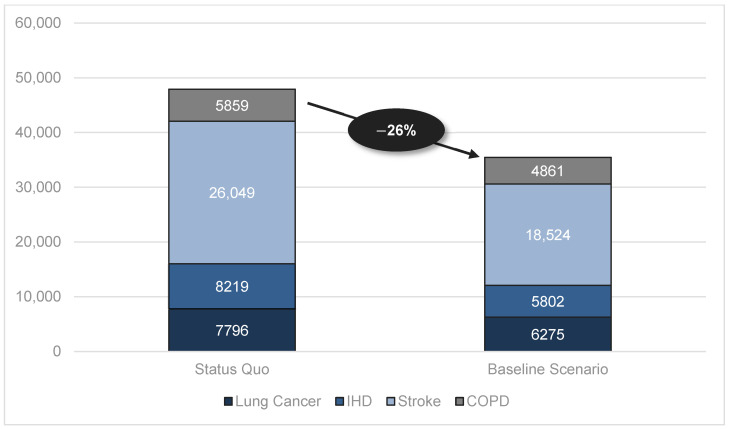
Number of smoking-attributable patients (in thousands).

**Figure 2 healthcare-12-01937-f002:**
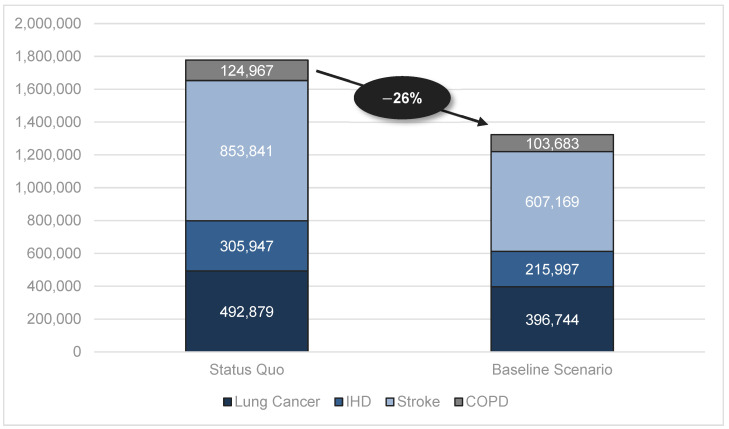
Number of smoking-attributable health care costs (in million JPY).

**Figure 3 healthcare-12-01937-f003:**
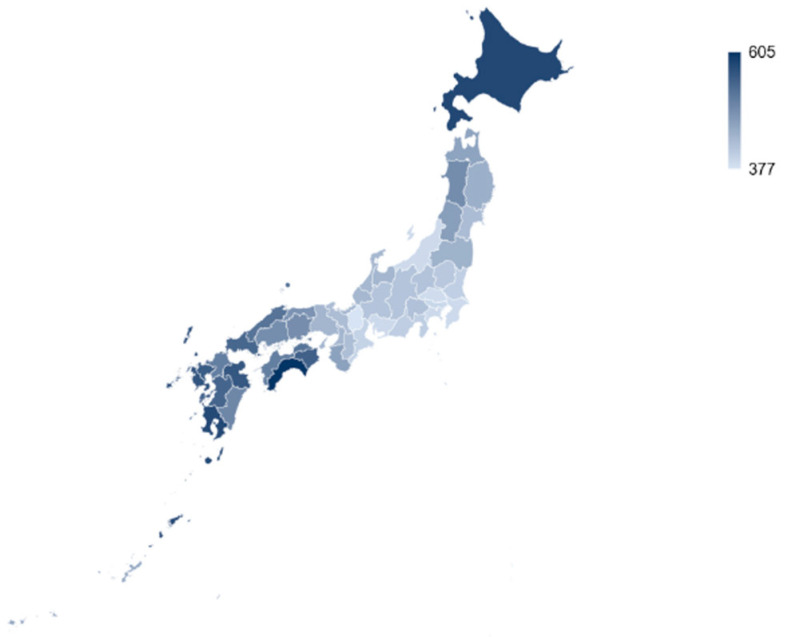
Savings in health costs by prefecture (in million JPY per 100,000 people).

**Figure 4 healthcare-12-01937-f004:**
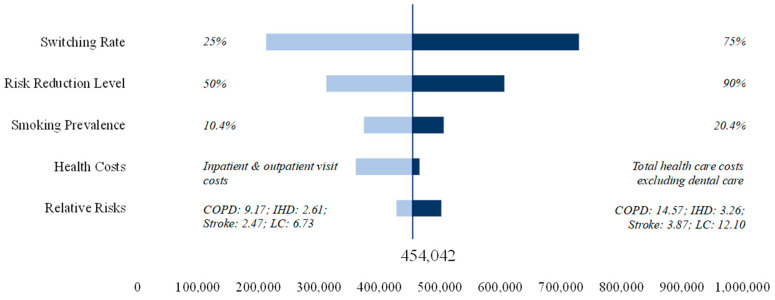
Sensitivity analysis for health care cost savings (million JPY).

**Table 1 healthcare-12-01937-t001:** Modeling assumptions and data input for baseline scenario.

	Assumptions	Source
Number of smoking-attributable patients in status quo	47,922,652	Own calculations based on [3,23,39,40,42,48]
COPD	5,858,979	
IHD	8,218,702	
Stroke	26,049,067	
Lung cancer	7,795,904	
Relative risks of current smokers		[39,40]
COPD	11.57	
IHD	2.91	
Stroke	3.12	
Lung cancer	8.96	
Annual smoking-attributable health care costs in status quo in million JPY	1,777,635	Own calculations based on [3,23,39,40,42,48]
COPD	124,967	
IHD	305,947	
Stroke	853,841	
Lung cancer	492,879	
Risk reduction after switching to HTP	70%	[49]
Smoking prevalence	15.4%	[23,42]
Switching rate	50%	[25]

COPD: chronic obstructive pulmonary disease; IHD: ischemic heart disease; HTP: heated tobacco product; JPY: Japanese yen (1 JPY = 0.0066 U.S. dollar as of April 2024).

**Table 2 healthcare-12-01937-t002:** Assumptions for the deterministic sensitivity analysis.

	Lower End		Upper End	
	Assumption	Reference	Assumption	Reference
**Risk Reduction Level**	50%	[36]	90%	[30,50]
**Relative Risks**				
**COPD**	9.17	[39]	14.57	[39]
**IHD**	2.61	[39]	3.26	[39]
**Stroke**	2.47	[39]	3.87	[39]
**Lung cancer**	6.73	[40]	12.10	[40]
**Switching Rate**	25%	[25]	75%	[25]
**Smoking Prevalence**	10.4%	[23,42]	20.4%	[23,42]
**Health Costs**	Inpatient and outpatient visit costs	[48]	Total health care costs excluding dental medical care	[48]

**Table 3 healthcare-12-01937-t003:** Number of smoking-attributable patients (in thousands).

	Status Quo	Baseline Scenario	Δ	In%
Total	47,923	35,462	12,460	−26%
Lung Cancer	7796	6275	1521	−20%
Ischemic Heart Disease (IHD)	8219	5802	2416	−29%
Stroke	26,049	18,524	7525	−29%
Chronic Obstructive Pulmonary Disease (COPD)	5859	4861	998	−17%

**Table 4 healthcare-12-01937-t004:** Smoking-attributable health care costs (in million JPY).

	Status Quo	Baseline Scenario	Δ	In%
Total	1,777,635	1,323,593	454,042	−26%
Lung Cancer	492,879	396,744	96,135	−20%
Ischemic Heart Disease (IHD)	305,947	215,997	89,951	−29%
Stroke	853,841	607,169	246,672	−29%
Chronic Obstructive Pulmonary Disease (COPD)	124,967	103,683	21,284	−17%

## Data Availability

The original contributions presented in the study are included in the article, further inquiries can be directed to the corresponding author/s.

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
