# Peer review of "Switching from Cigarettes to Heated Tobacco Products in Japan—Potential Impact on Health Outcomes and Associated Health Care Costs"

_healthcare, 2024, doi:10.3390/healthcare12191937_

Round 1

Reviewer 1 Report

Comments and Suggestions for Authors

The paper explores the potential impact of switching from cigarettes to heated tobacco products on health outcomes and public health costs in Japan. It’s a relevant and timely topic that contributes to the current discussion.

My main feedback focuses on improving the robustness of the estimates, which is the foundation for any conclusion drawn in this paper:

1.      The paper uses smoking prevalence rates from NHNS 2016 and extrapolates these based on 2019 NHNS data. It would be helpful to explain how this extrapolation is performed.

2.      To make it clear, it would be beneficial to include the mathematical equations of the entire model. This would show how each part of the estimation contributes to the final figures, making the calculations more transparent and easier to verify.

3.      While the authors have clearly put significant effort into the estimations, presenting some validation against baseline scenarios would be necessary. Comparing these figures with estimates from other sources (e.g., IHME) could provide additional support.

4.      The model currently assumes no changes in behavior due to switching to heated tobacco products. It’s important to consider that such a switch might affect the prevalence and duration of use. A sensitivity analysis to explore this assumption could make the estimates more realistic.

Reviewer 2 Report

Comments and Suggestions for Authors

The article is dedicated to discovering health care system privileges from switching from cigarettes to less harmful heated products for a population of Japan. The authors support the article with appropriate statistical material and analysis of tendencies in the healthcare sector related to cigarette-tailored disease occurrence among different aged people. The paper is interesting from a practical point of view, mainly because of its valuable implications for implementing the state government policy. This paper encompasses deep theoretical background analysis in the form of a literature review and results of an investigation, which, in combination, allow for a considerable Discussion chapter that draws the reader's attention.

However, I would like to say about some drawbacks of the manuscript:

1)    Please unify all references in the text according to the journal's requirements.

2)    Check the manuscript so that all statistical material must have appropriate citations in the Reference list.

3)    Pay more attention to describing possible advantages of HTPs because with the wide spreading of the idea of a healthy lifestyle among citizens of different countries, switching to alternative (less, but harmful too) types of products is not easy to incorporate into the modern concept of healthy well-being.

4)    Lines 274-281 this information is too obvious to pay attention to here. I recommend enriching the Policy Implication Section with the authors’ additional recommendations.

5)    Enlarge Conclusions and describe the principal novelty of your research. Compare its findings with previous research in this field. What are the prospects of this investigation?

Reviewer 3 Report

Comments and Suggestions for Authors

1)     The abstract section has not been prepared following the journal's requirements. Accordingly, revise it.

2)     Avoid using abbreviations in the abstract section because there are many abbreviations in the abstract section in the current form.

3)     Abbreviations must be written in explicit form at the first occurrence. For instance, JPY, OECD.

4)     The primary motivation of the student has not been successfully uncovered.

5)     Which study could fill the literature gap is not provided.

6)     The theoretical discussions were weak for a paper intended for publication in a high-quality journal.

7)     The former literature is not critically examined in the study.

8)     The organization and flow of the study were not constructed to be reader-friendly.

9)     The policy recommendation section lacked the required soundness and innovative proposals.

10)  The authors should visualize the numbers in the tables to provide insights for the readers at a glance.

Comments on the Quality of English Language

Minor editing of English language required.

Round 2

Reviewer 2 Report

Comments and Suggestions for Authors

Could be accepted in the present form

Author Response

No more open requests

Reviewer 3 Report

Comments and Suggestions for Authors

The authors have made significant efforts for addressing former queries, and they have done good job. However, the literature gap is still insufficiently discussed. Therefore, there is a need for one more round of minor revision in the study.

Author Response

The authors have made significant efforts for addressing former queries, and they have done good job. However, the literature gap is still insufficiently discussed. Therefore, there is a need for one more round of minor revision in the study.
response:

we have added the following paragraph to highlight the literature gap: „The objective of this study is to address the existing literature gap concerning the macroeconomic consequences of the health implications of HTPs. The majority of current research on HTPs is concerned with the health effects of consumption at the individual level, with a particular focus on the behavioral and physiological impacts at the micro level [32-40, 49,50]. However, there has been comparatively little exploration into the influence of these estimated health outcomes on macroeconomic factors. A modest but growing body of research has begun to address this issue, with similar simulations conducted for theUnited States (US) and Mexico for e-cigarettes [[i],[ii]]. This study will be the first to apply such simulations in the context of Japan, as well as the first to do so specifically for HTPs. Notably, related simulations of tobacco interventions in Japan, such as those modeling the impact of varenicline for smoking cessation [14], will be used for comparison in our analysis.

Japan’s high smoking prevalence, combined with its widespread acceptance of smoke-free alternatives, presents a unique opportunity to investigate whether transitioning smokers to heated tobacco products (HTPs) could alleviate the burden on the healthcare system. We seek to address the question: Can risk-reduced products serve as a viable alternative for mitigating the health burden associated with smoking?

[1] Levy, D. T., Borland, R., Lindblom, E. N., Goniewicz, M. L., Meza, R., Holford, T. R., ... & Abrams, D. B.  Potential deaths averted in USA by replacing cigarettes with e-cigarettes. Tobacco control2018; 27(1), 18-25.

[1] Sánchez-Romero, L. M., Li, Y., Zavala-Arciniega, L., Gallegos-Carrillo, K., Thrasher, J. F., Meza, R., & Levy, D. T. The potential impact of removing a ban on electronic nicotine delivery systems using the Mexico smoking and vaping model (SAVM). medRxiv : the preprint server for health sciences, 2024.04.28.24306511. https://doi.org/10.1101/2024.04.28.24306511
